# Quality of Life and Incidence of Clinical Signs and Symptoms among Caregivers of Persons with Mental Disorders: A Cross-Sectional Study

**DOI:** 10.3390/healthcare12020269

**Published:** 2024-01-20

**Authors:** Vasiliki Oikonomou, Evgenia Gkintoni, Constantinos Halkiopoulos, Evangelos C. Karademas

**Affiliations:** 1School of Social Sciences, Hellenic Open University, 26335 Patras, Greece; v_oikonomou@windowslive.com (V.O.); karademas@uoc.gr (E.C.K.); 2Department of Psychiatry, University General Hospital of Patras, 26504 Patras, Greece; 3Department of Management Science and Technology, University of Patras, 26334 Patras, Greece; halkion@upatras.gr; 4Department of Psychology, University of Crete, 74100 Rethymnon, Greece

**Keywords:** clinical symptoms, caregivers, mental disorders, quality of life, stress, anxiety, depression, social support

## Abstract

Background: Caring for individuals with mental disorders poses significant challenges for caregivers, often leading to compromised quality of life and mental health issues such as stress, anxiety, and depression. This study aims to assess the extent of these challenges among caregivers in Greece, identifying which demographic factors influence their well-being. Method: A total of 157 caregivers were surveyed using the SF-12 Health Survey for quality-of-life assessment and the DASS-21 questionnaire for evaluating stress, anxiety, and depression symptoms. *t*-tests, Kruskal–Wallis tests, Pearson’s correlation coefficients, and regression analyses were applied to understand the associations between demographics, quality of life, and mental health outcomes. Results: The study found that caregivers, especially women and younger individuals, faced high levels of mental health challenges. Marital status, educational level, and employment status also significantly influenced caregivers’ well-being. Depression was the most significant factor negatively correlating with the mental component of quality of life. The magnitude of the burden experienced by caregivers highlighted the urgency for targeted social and financial support, as well as strategic treatment programs that consider caregiver well-being. Conclusions: Caregivers of individuals with mental disorders endure significant stress, anxiety, and depression, influencing their quality of life. Demographic factors such as age, gender, marital status, education, and employment status have notable impacts. Findings emphasize the need for society-wide recognition of caregivers’ roles and the creation of comprehensive support and intervention programs to alleviate their burden, particularly in the context of the COVID-19 pandemic.

## 1. Introduction

Mental disorder has extensive repercussions for both patients and their caretakers. The burden of caregiving has been evaluated in the literature using studies whose criteria vary. However, there is widespread agreement that caregivers are heavily burdened by their demanding daily lives and are frequently unsupported in their caring role. Emotional burden and burnout can negatively impact the mental health and quality of life of mental disorder caregivers and worsen the clinical course of their patients.

The majority of previous research has focused on caregivers of individuals with long-term physical illnesses, resulting in a significant lack of knowledge regarding the experiences of those who care for individuals with mental disorders. Comparable research can shed light on the distinct difficulties encountered by these caretakers. It is essential to examine the correlation between providing care for individuals with mental health disorders and the prevalence of depressive and anxiety symptoms in caregivers.

For example, according to research [1], long-term care caregivers may significantly reduce the likelihood of increased hospitalizations for people with mental disorders. Through this research, it is essential to understand how caregivers of people with mental disorders experience anxiety and stress due to the long-term care they provide, their quality of life, and their interactions with the healthcare system [2].

Undoubtedly, the care provided by caregivers to individuals may also impact their own mental well-being. Additional research can highlight the direct impact of caregiving on mental well-being, underscoring the need for mental health services and interventions specifically targeted towards caregivers. The findings obtained from research carried out on caregivers could be pivotal in influencing healthcare policies and practices designed to enhance the mental well-being of this specific subgroup [3]. They have the ability to guide the progress of healthcare models that are more comprehensive, placing patients at the forefront and taking into account the welfare of caregivers. By recognizing the difficulties faced by these caregivers, such studies can help to empower them. It can enhance awareness regarding their needs and facilitate the development of resources and networks that provide the necessary assistance [4].

Examining the well-being of caregivers of individuals with mental disorders and the prevalence of clinical signs and symptoms is essential because it dramatically affects the caregivers’ quality of life and highlights the necessity for comprehensive support initiatives. The caregivers of individuals with mental disorders frequently encounter substantial levels of stress, anxiety, and depression, which can significantly impact their overall well-being [4]. It is essential to comprehend the influence of caregiving on the well-being and psychological state of caregivers of individuals with mental disorders. This understanding is vital for creating specific interventions and support systems catering to their needs and difficulties.

Additionally, studies conducted in Greece have assessed caregivers’ burden on people with chronic diseases; however, the caregiver experience of people with mental disorders has received scant attention [5]. Therefore, it is anticipated that this article will significantly contribute to recognizing the difficulties caregivers of people with mental disorders face and will encourage the development of educational programs to support and empower them. Furthermore, this study seeks to investigate the relationship between the quality of life of mental disorder caregivers and the occurrence of depressive and anxiety symptoms [6].

In recent years, caregivers’ perceptions of health and quality of life have been the subject of extensive research. This study will concentrate on the quality of life of mental illness caregivers so that this issue can be better documented for this particular group of caregivers. Because in recent years, a significant portion of the literature has focused on the caregivers of people with chronic diseases, the collection of information about the quality of life and the incidence of depressive and anxiety symptoms of caregivers of people with a mental disorder will provide meaningful data for this specific group of caregivers.

The purpose of this work is to investigate the quality of life of caregivers of people with a mental disorder, as well as its correlation with the occurrence of depressive and anxiety symptoms. For the implementation and design of this research protocol, the following research questions were formulated:(RQ1) In which domain (physical/physical or mental) is there a more significant burden?(RQ2) Which demographic factors are significantly associated with the level of quality of life of caregivers of people with a mental disorder?(RQ3) Is there a significant correlation between the levels of depressive and anxiety symptoms and the level of quality of life of caregivers of people with a mental disorder?

## 2. Literature Review

### 2.1. Quality of Life of Caregivers

Quality of life is essential for modern societies and economies, but it remains a profoundly anthropocentric and complex concept because it attempts to link human needs and activities. Everyone’s interpretation of the term quality of life is highly subjective. From a scientific and social perspective, quality of life comprises multiple variables and factors contributing to a quality whole. Quality of life is a blueprint for physical health, psychological parameters, and social interaction [3,4]. The components of physical health are mobility, independence, the capacity to work, and the optimal operation of vital organs. It manifests itself in family life, romantic relationships, and other social manifestations in the social aspect of quality of life. This term became more prevalent in the 1980s as an indicator of social research aimed at eliminating inequalities and ensuring the equitable distribution of resources.

According to the research, physical and mental stress among caregivers contributes to a decline in their quality of life. It has been established that the reactions and consequences of caregiving make the role of caregiver challenging and, consequently, a source of anxiety [5]. Schizophrenia caregivers report a lower quality of life than non-caregivers and caregivers of people with other illnesses [6]. The objective and subjective burden of informal caregiving is substantial, resulting in diminished quality of life [7]. Reports that 38.2% of mental disorder caregivers experience a severe burden in their caregiving role. Caregivers of individuals with a mental disorder spend an average of 22 or more hours per week providing care. Some researchers [8] found a significant correlation between caregiver burden and quality of life.

According to another study [9], caregiver age, gender, education level, job loss due to caregiving demands, income, relationship with the patient, frequency of caregiving, and duration of patient illness are significant correlates of caregiving burden. In addition, the severity of the patient’s symptomatology, its duration, the number of needs, the extent of the disability, and the diminished social interests compound the burden for mental disorder caregivers [10,11].

Low burden of care and social and professional support have been identified as positive predictors of the quality of life of caregivers of individuals with mental disorders [8]. Caregivers believe their quality of life would improve if they had more time for themselves and maintained a greater distance from the individual they are tending to. In addition, the caregiver’s independence is essential to pursue interests and activities interrupted by caregiving [12]. Special consideration has been given to the mental health of caregivers and the difficulties they may experience, such as anxiety, depression, and distress, since their caregiving responsibilities present several obstacles [13].

### 2.2. Anxiety, Depression, and Burden in Caregivers

Family members with a close relationship with the patient provide more extended care, devoting most of their time to caregiving, a significant source of stress and burden. In comparison to caregivers who did not receive any educational support, burden, and depression symptoms are significantly lower among those who have received an educational intervention for standard care [14,15,16]. Within three months, caregivers who received training experienced a significant reduction in depressive symptoms, from 36% to 17%. In contrast, the prevalence of depressive symptoms among untrained caregivers increased from 22% to 50% over the same time frame [17].

It is true that some caregivers’ feelings of isolation and helplessness can hinder their ability to provide quality care, limit their goals and life activities, diminish their quality of life, and contribute to mental health issues such as anxiety and depression [18]. The experience of providing care is multifaceted and intricate. Caregivers play a crucial role in assisting individuals with severe mental illness. Supporting caregivers by reducing their burden and enhancing their mental health enables them to continue providing care for their loved ones despite the difficulties of caregiving. Educating caregivers can alleviate their responsibilities and improve their quality of life.

Informal care provides daily health care to dependent individuals by family members, friends, neighbors, or anyone else in the immediate social network. However, they typically do not receive monetary compensation for their assistance [19]. Typically, informal caregivers are members of a close-knit family; however, it is sometimes observed that this responsibility is fragmented and shared by multiple individuals. In order to meet this new challenge, families who primarily provide informal care develop coping mechanisms for their increased stress levels. These include passive coping, reframe, spiritual, social, and health service support [20].

In recent years, caregiver burden and the difficulties of providing care have been the subject of growing research. There are numerous ways to present caregiver burden. However, the terms objective and subjective burden are the most prevalent. Objective burden represents periods of absence from caregiving that are observable and verifiable, whereas subjective burden represents personal feelings of burden [21].

Numerous studies have demonstrated that caregivers experience elevated levels of anxiety and depression. For instance, caregivers of people with schizophrenia are likelier than those with other chronic illnesses to experience sleep difficulties, insomnia, pain, and anxiety [5]. The prevalence of depression in long-term caregivers is 15–32%, 1–10% higher than in the general population. Due to a lack of understanding of the disease and the non-use or absence of social support systems, the incidence of depressive symptoms among family members increases immediately following a disease diagnosis [22,23]. In research conducted in 2018 [17] in Japan, the rate of depression among long-term dementia caregivers before attending a training program was 36%, while in the United Kingdom, it was 29.4% [24].

Moreover, it has been observed that caregivers of individuals with dementia often encounter feelings of anxiety, as emphasized in a comprehensive analysis [25]. The study revealed a positive correlation between caregiver burden and poorer caregiver physical health with heightened levels of anxiety. These factors were additionally correlated with an elevated susceptibility to depression. Furthermore, another study [26] discovered that caregivers of children with asthma who had a low family socioeconomic status, low-income family functioning, and a heightened tendency to experience shame exhibited elevated levels of anxiety and depression.

Numerous academic studies [25,26,27,28,29] have examined the correlation between mental disorders, specifically anxiety and depression, and the consequential influence on the caregivers’ quality of life. In general, the available evidence indicates that mental disorders, including anxiety and depression, have a detrimental effect on the quality of life experienced by caregivers. Various factors, including caregiver burden, physical health, and family functioning, can potentially contribute to the onset and progression of mental disorders. Acknowledging and resolving caregivers’ mental health requirements are imperative for enhancing their overall welfare and the standard of care they deliver. In a specific investigation [28], a correlation was discovered between the emotional distress experienced by spousal caregivers, such as symptoms of depression and anxiety, and the diminished quality of life observed in the patients they care for.

## 3. Materials and Methods

### 3.1. Participants

The sample of participants in the present study includes:All caregivers of people with a mental disorder who visit the psychiatric department;The outpatient departments of the psychiatric department;The psychiatric emergency department of a General Hospital in Greece.

The study’s sample comprises all caregivers of individuals with a mental disorder who receive therapy at the psychiatric department, outpatient clinics, and psychiatric emergency department of the General Hospital of the Peloponnese Region in Greece. The characteristics of the sample are described in more detail in the findings/results of this research. The inclusion criteria for the study were the following: (a) consent to participate in the research and (b) being caregivers (close relatives) of people with a mental disorder.

This study was carried out at a General Hospital from October to December 2022. The health unit’s selection criteria consisted of the everyday duties of the hospital’s psychiatric clinic. The study included the outpatient clinics of the psychiatric clinic and the emergency department dedicated to addressing psychiatric manifestations.

### 3.2. Research Hypotheses

The research hypotheses formulated are that: (RH1) there is a difference in the burden of caregivers of people with a mental disorder in the physical/physical sector and the mental sector; (RH2) demographic factors have an effect on the occurrence of depressive and anxiety symptoms in the caregivers of people with a mental disorder; (RH3) there is a significant correlation between the levels of depressive and anxiety symptoms and the level of quality of life of the caregivers of people with a mental disorder. For this research, a quantitative methodology was implemented. The quantitative method permits an objective evaluation of a phenomenon (in this case, the quality of life and depressive and anxiety symptoms). In addition, the time horizon of the study is a crucial research parameter. There are two primary approaches: cross-sectional and longitudinal studies. It was decided to conduct a cross-sectional study due to time constraints and the need to assess the quality of life and the occurrence of depressive and anxiety symptoms in a specific time frame for a specific group of caregivers (caregivers of individuals with a mental disorder). As indicated in another study [30], note that the most significant advantage of these studies is that they are generally fast and reliable; this methodology was utilized in the present study.

Furthermore, the most effective approach to establish connections between variables (specifically, the relationship between quality of life and symptoms of depression or anxiety) is to carry out cross-sectional studies. Given these circumstances, conducting such research is essential for the ongoing investigation.

### 3.3. Diagnostic Assessments

As a research instrument, a questionnaire with closed-ended questions was utilized and administered by the primary researcher in an in-person interview. The questionnaire comprises three sections in total. The first section of the survey documented the caregivers’ demographic information (gender, age, level of education, marital status, income, and occupation).

In the second section, the patient’s quality of life was evaluated utilizing the SF-12 Health Survey [31,32], and in the third section, the patient’s anxiety and depression levels were evaluated utilizing the DASS-21 [33,34].

### 3.4. Description of Psychometric Scale SF12

The SF-12 questionnaire was developed as a short version of the original SF-36 for large-scale studies, and its reliability and validity have been established [31,32]. In Greek, some researchers [35] standardized the SF-36 and SF-12 instruments. The SF-12 consists of 12 questions in total. The 12 queries assess the patient’s physical (PCS) and mental health (MCS). Higher values indicate a higher health-related quality of life, while the mean value is 50 with a standard deviation of 10. Based on the PCS and MCS scales, the SF-12 provides a brief assessment of the patient’s health-related quality of life, for which it is widely acknowledged scientifically. Together, mental and physical health assess subscales of physical functioning, vitality, social functioning, general health, physical pain, mental health, and limitations in social functioning due to emotional problems [35,36,37,38].

### 3.5. Description of Psychometric Scale DASS-21

The DASS-21 was developed as a short form of the DASS-42 and has been reported to have slightly improved psychometric properties compared to the entire DASS [39]. The DASS-21 is a 21-item self-report scale that measures levels of depression, stress, and anxiety in the population. Each seven (7)-item scale has four (4) response options ranging from zero (0) (does not apply to me at all) to three (3) (applies to me a lot or most of the time). Its maximum score is 42. The DASS-21 scale score has excellent internal consistency, and interpretive scores have good construct validity [40,41,42,43].

Standardization of the DASS-21 involves the assessment of its psychometric properties, including reliability (consistency of results over time) and validity (accuracy in measuring what it is supposed to measure). The Cronbach’s alpha coefficients, a measure of internal consistency, are typically high for all three subscales, indicating that the items within each subscale reliably measure the same underlying construct. Also, the DASS-21 has been validated against other established measures of depression and anxiety, showing good convergent and discriminant validity [44]. It is effective in differentiating between the symptoms of depression, anxiety, and stress, which often overlap in other measures.

### 3.6. Analysis

Data analysis was performed using SPSS v25 software. The prevalence of anxiety, depression, and caregivers’ quality of life will be described using descriptive indicators (frequency and percentage or mean value and standard deviation). Using the *t*-test of independence and *t*-test for equality of two means, as well as logistic regression, we investigated the relationship between anxious and depressive symptoms and the risk of low quality of life among caregivers of persons with mental disorders. Using the *t*-test of independence and multiple regression models, the effect of demographic and clinical characteristics of patients on their quality of life and risk of developing depressive and anxiety symptoms was investigated. Lastly, Pearson’s correlation coefficient and multiple regression models were used to examine the relationship between the quality of life and the level of anxiety and depression.

## 4. Results

### 4.1. Descriptive Data and Preliminary Analyses

The survey was conducted between October and December 2022 and was answered by caregivers (N = 157) of people with a mental disorder. The following results (Table 1) have been extracted using the SPSS statistical package.

Table 1 presents the demographics of the sample. In particular, of the N = 157 people, the majority (68.8%, N = 108) were women, and 30.6% (N = 48) were men. Regarding their educational level, most (37.6%) are university graduates. Regarding their work, 40.1% are employed in the private sector, 24.8% are civil servants, 12.7% are domestic workers, 11.5% are retired, 6.4% are self-employed, and, finally, 4.5% are unemployed.

### 4.2. Caregiver Quality of Life Analysis

In this section, an attempt will be made to analyze the quality of life of caregivers. Initially, the reliability of the questionnaire was checked through Cronbach’s alpha coefficient using the SPSS statistical program. For the Physical Health subscale (PCS), Cronbach’s alpha index was found to be 0.825, while for the Mental Health subscale (MCS), the index equals 0.824. Since values above 0.7 are considered acceptable, it is judged that the analysis shows reliability. Table 2 presents the basic descriptive statistics of the individual questions of the SF-12 questionnaire. In particular, Table 2 presents the mean, standard deviation, median, and frequency of the respondents’ answers, which is confirmed by the corresponding graphs. Higher mean values also indicate a higher level of quality of life.

The following section of the analysis analyzes the PCS and MCS subscales to determine the domain with the more significant burden. The descriptive statistics of caregiver responses to the PCS and MCS subscales for the physical and mental health of the SF-12 are presented in Table 3. Physical health has a mean of 48.68 and a standard deviation of 8.41, whereas mental health has a mean of 35.37 and a standard deviation of 10.73.

Nonetheless, the most crucial aspect of the analysis is to identify the demographic factors that may influence the two domains of the quality of life of mental disorder caregivers. The subsequent step of the analysis investigates whether the sample’s demographic characteristics influence the mean quality-of-life scores. At this juncture, the following inspections will be conducted:

Regarding gender, the independent sample *t*-test’s parametric control will be computed because the variable is dichotomous and there are more than 30 observations in each category. The null hypothesis is that the mean values are equal, and the alternative hypothesis is that they differ. This control does not include the observation regarding the individual who selected “Other”, as a single observation cannot account for any difference between the means.

Regarding the demographic factors of age, income, marital status, educational level, and employment, the non-parametric Kruskal–Wallis test is used because there are more than 2 categories and fewer than 30 observations per category. The null hypothesis is that the variables in question are independent, while the alternative is that they are dependent. Despite the initial preference for the ANOVA test, it was discovered that some subcategories of the variables contained a small number of observations. Therefore, the Kruskal–Wallis test, which is more powerful in detecting differences in this data pattern, was favored. The significance level in all tests was set at 5%. Therefore, the original hypothesis is accepted when the *p*-value (*p*) ≥ 0.05 and rejected when the *p*-value (*p*) < 0.05. For the case of gender, a statistically significant difference according to gender is found only in terms of the mental component (MCS) (Table 4).

The next analysis stage examines the possible differentiation based on the Kruskal–Wallis test that presents the relevant results, from which the following emerges (Table 5):There is a statistically significant difference by age for the PCS component;There is a statistically significant difference in both factors of the SF-12 scale depending on the respondents’ marital status, educational level, and employment;There is a statistically significant difference in income for the PCS component.

#### 4.2.1. Assessment of Anxiety and Depression Level

Table 6 presents the descriptive statistics for each questionnaire question and the basic descriptive statistics of the individual questions of the DASS-21 questionnaire. Also, Table 6 presents the mean, the standard deviation, the median, and the frequency of the responses given by the respondents, which is also confirmed by the corresponding graphs.

It should be mentioned that higher mean values also indicate a higher level of stress/anxiety/depression. As can be seen, the symptoms that have a more significant effect on these variables are difficulty in relaxing, the tendency of the respondents to overreact to the situations they faced, and feelings of discomfort, irritability, nervousness, and frustration, which are also confirmed by the relatively higher averages compared to the rest of the questions.

The analysis continues with the descriptive statistics of the three subscales of the questionnaire, as presented in Table 7. First, regarding the stress scale, it is observed that the respondents show mild levels of stress with a mean value equal to 14.87 and a standard deviation of 9.08 units. Next, examining the anxiety scale, it appears that the mean value of anxiety equals 7.38 with a standard deviation of 7.36 points. This value again indicates a mild level of stress for the respondents. Finally, checking the depression scale, it is observed that normal levels of depression characterize the respondents with a mean value of 9.36 and a standard deviation of 8.46 units. Therefore, it could be said that there does not seem to be a very high level of anxiety and depression in the sample.

It is then examined whether there is a statistically significant relationship between demographic factors and the values of the individual scales of the DASS-21 questionnaire. Table 8 presents the *t*-test for the case of gender. In this case, a statistically significant difference according to gender is found only in terms of the mental component (MCS).

The next analysis stage examines the possible differentiation based on the Kruskal–Wallis test. Table 9 presents the relevant results, from which the following emerges:There is a statistically significant difference in age regarding anxiety and depression parameters;There is a statistically significant difference in terms of all three parameters of the DASS-21 scale depending on marital status and educational level;There is a statistically significant difference depending on employment regarding the stress parameter;There is no statistically significant difference with any parameter according to income.

#### 4.2.2. Correlation of Depressive and Anxiety Symptoms of Caregivers’ Quality of Life

The last stage of the analysis examines the correlation found between the two subscales of the SF-12 questionnaire and the three categories of the DASS-21 questionnaire. At the first level, the Pearson’s correlation coefficient is examined. Table 10 presents the correlation coefficient between the subscales of the SF-12 questionnaire and the variables resulting from the analysis of the DASS-21 questionnaire. The analysis showed the following:Regarding the physical factor (PCS), there was a negative correlation with all three categories of the DASS-21 questionnaire, which is statistically significant at the 1% level for the stress and depression variables and the 5% level for the anxiety variable. However, the correlation is weak since the correlation coefficients are small.Regarding the mental factor (MCS), there was a statistically significant negative correlation with all three categories at the 1% significance level.Finally, it was found that the three variables of the DASS-21 scale are positively correlated, which shows that the increase or decrease in one of the three factors can lead to a corresponding increase or decrease in another factor.

## 5. Discussion

In this study, the quality of life of mental disorder caregivers and their correlation with depressive and anxious symptoms were examined. Additionally, the research questions (RQs) that were posed in the study are presented and indicated in this section. One hundred fifty-seven caregivers of individuals with mental disorders who visited the mental health department of a General Hospital participated in the study. Regarding the participant profile, 68.8% were female, 30.6% were male, and 0.6% identified as gender neutral. According to research [45], women are twice as likely as males to fulfill this informal obligation for their family members. Other studies have found the same, attributing the responsibility of care primarily to women, daughters, and spouses, either because of their deep-seated belief that this role is theirs or because of the social environment.

Based on similar research, it has been observed that caregivers of individuals with mental disorders may encounter notable impacts on their quality of life [46]. These effects may manifest in diminished quality of life resulting from heightened psychological strain, compromised physical well-being, and socioeconomic difficulties. Numerous scholarly investigations have been conducted to assess caregiving’s influence on caregivers’ overall well-being, yielding consistent empirical evidence in support of this assertion. A study conducted amidst the COVID-19 pandemic revealed a decline in care providers’ physical and mental well-being. Another study [47] also emphasized the significant impact of social isolation on various dimensions of caregivers’ quality of life, such as heightened levels of depression and anxiety. Other researchers conducted a study [48] that examined the relationship between the mental and physical health of advanced cancer patients and their family caregivers, revealing a potential interdependence between the two parties.

According to the SF-12 questionnaire, the quality of life of caregivers of individuals with mental disorders was analyzed in the two main categories of physical/somatic health and mental health, along with their subcategories. For example, energy, serenity, and irritability/depression have an intermediate impact on the respondents’ quality of life. (RQ1) In contrast, a substantial proportion of respondents assert that their life quality is unaffected by factors corresponding to their firm belief regarding the general state of health.

Furthermore, a research investigation in England yielded compelling evidence indicating that the provision of care significantly impacts the mental well-being of those who assume the caregiving role [49]. The caregivers of individuals with mental illness experience a significant burden, which can have detrimental effects on both their quality of life and the well-being of those they care for [50]. In summary, a multitude of studies have yielded empirical evidence that substantiates the assertion that the well-being of caregivers for individuals with mental disorders may be substantially impacted. Caregivers may encounter heightened psychological distress, physical health concerns, and socioeconomic difficulties. The reciprocal relationship between patients’ and caregivers’ mental and physical health has been observed in diverse contexts.

According to the findings of the present research, participants reported more concerns with mental health compared to physical health. Caregivers of people with mental disorders appear to have poorer mental health than caretakers of people with chronic illnesses and non-carers [49,51] (RQ1).

Numerous studies have been conducted to investigate the implications of caregiving on the mental well-being of individuals, yielding consistent and compelling evidence supporting this assertion. Significant mental health burden was observed among caregivers of individuals diagnosed with mental disorders, such as schizophrenia and anorexia nervosa. The act of providing care was found to be linked with increased levels of depression, anxiety, and stress in caregivers [52,53]. Moreover, the ongoing COVID-19 pandemic has further intensified the mental health challenges faced by individuals who provide care to others. Research has indicated that individuals responsible for the care of those with mental disorders encountered mental health challenges amidst the pandemic [54,55]. The ongoing global pandemic has resulted in a notable escalation of stress, anxiety, and depression among individuals who provide care, thus underscoring the imperative for supplementary assistance and available resources. Caregivers may encounter heightened levels of stress, anxiety, and emotional burdens concerning the ongoing provision of care and support. The responsibility of providing care can potentially contribute to the development of psychiatric disorders and have adverse effects on the overall well-being of individuals fulfilling the caregiving role.

According to the findings, the caregivers of persons with mental disorders who participated in this study have moderate stress and anxiety levels, with average values of 14.87 and 7.38 (RQ3). In terms of depression, the respondents have normal levels. In contrast, the findings of a study [56] indicated that the highest proportion of caregivers of individuals with chronic diseases exhibited depressive symptoms due to high psychological strain. In terms of demographic characteristics, female caregivers exhibit higher levels of tension, anxiety, and depression symptoms than male caregivers and younger caregivers than older ones. Moreover, those who have lost their partner, i.e., are widowed and are engaged in housework, experience more significant stress, anxiety, and depression than those who are married and employed, who experience less clinical signs. (RQ2).

Numerous academic studies have explored the correlation between caregiving responsibilities and the subsequent impact on the mental well-being of caregivers, thereby furnishing substantiating evidence in favor of this assertion. Previous studies have indicated that individuals who provide care for those with mental disorders, such as schizophrenia, bipolar affective disorder, and substance use disorder, tend to exhibit elevated levels of depressive and anxiety symptoms [57,58,59,60]. The combination of the responsibility of providing care and the societal disapproval linked to specific mental disorders can lead to heightened depressive symptoms among individuals who assume the role of caregivers [61]. According to a study [62], caregivers of individuals with dementia were observed to exhibit psychopathological symptoms, such as depression and anxiety. There is a consistent association between the intensity of the care recipient’s symptoms and elevated levels of caregiver strain and mental health symptoms, as indicated by multiple studies [63,64,65]. Several research studies conducted in Sri Lanka and the Netherlands have revealed that individuals who provide care for individuals with mental disorders tend to experience elevated levels of depressive symptoms and heightened caregiver strain [66,67]. In a similar vein, research conducted in Brazil and Spain has revealed that caregivers of individuals with mental disorders demonstrated elevated levels of emotion expressed and experienced significant levels of stress [68,69].

In an attempt to determine whether the demographic characteristics of the participants influence the two domains of physical and mental health of caregivers of people with mental disorders, it was discovered that gender influences both domains of physical and mental health, with men reporting higher quality of life. (RQ2) The physical, mental, and general health status of informal female caregivers has been documented by a study [70] and is consistent with the results of the current study. Informal care is detrimental to the health of caregivers, who are predominantly women; 27.2% of female informal caregivers report health issues. In addition, the same researchers [70] have documented a correlation between a more significant perceived burden and poorer overall health in women, notably when social support is lacking.

Regarding age, it was discovered that participants of all ages experience superior physical health to mental health. Those with Greek citizenship also enjoy a superior quality of life than those without Greek citizenship. Regardless of the family circumstance, the level of physical health is consistently higher, indicating a higher quality of life.

Concerning the educational level of the caregivers of individuals with mental disorders, the graduates of university and college have a higher quality of life regarding the physical component. In comparison, secondary education graduates have a higher quality of life regarding their mental component. (RQ2) Researchers [71] argued that caregivers with a higher level of education are associated with a more significant burden, attributing it to the increased demands and expectations they may place on themselves and others. In contrast, in another study, researchers [72] and colleagues argue that the more education a caregiver has, the better equipped they are to handle the challenges of caregiving. In addition, a high level of education is associated with greater sociability, improved working conditions, higher pay, and increased family income, all of which contribute to a sense of security (RQ2).

Regarding employment, those engaged in domestic work and earning up to EUR 11,000 reported the highest quality of living in the physical domain. At the same time, retired caregivers enjoy a higher mental quality of life. In contrast, the perceived quality of life of informal caregivers of older citizens increased as their income increased, according to survey results. The DASS-21 self-administered questionnaire was used to evaluate the level of tension, anxiety, and depression. It was discovered that the factors that have a significant impact on the symptoms of stress, anxiety, and depression for caregivers of people with mental disorders are the inability to relax, the tendency to overreact to the situations they face, the feeling of discomfort, irritability, nervousness, and frustration. (RQ2).

The association between demographic factors of caregivers and mental health outcomes has also been documented in research examining caregivers of individuals with severe mental illness. Other researchers [73] discovered a correlation between the female gender and younger age of caregivers and an increased likelihood of experiencing mental distress. Moreover, the burden experienced by caregivers can be influenced by various factors, including the nature of the caregiver–patient relationship and cultural and ethnic variables [74]. The examination of the influence of demographic variables on the mental well-being of caregivers has also been explored within the framework of the COVID-19 pandemic. According to a study conducted by researchers [75], it was observed that female caregivers and older children experienced a greater degree of adverse effects on their mental well-being amidst the pandemic. The influence of socioeconomic factors, specifically lower socioeconomic status, has been recognized as a contributing element to the increased vulnerability of caregiver mental health [76].

The correlation between the two subscales of the SF-12 questionnaire (physical health and mental health) and the three categories of the DASS-21 questionnaire (stress, anxiety, and depression) was also examined. Regarding physical and mental health, it was discovered that there is a negative correlation with all three DASS-21 categories. (RQ3) This correlation is also supported by researchers [56], who discovered that caregivers with depressive symptoms report substantially lower levels of both the physical and mental aspects of quality of life. Caregivers of persons with a mental disorder report low levels of mental health by recording symptoms of anxiety, stress, and depression in percentages ranging from 30 to 40 percent, particularly in the first years following the diagnosis of the disorder and in early psychoses [51]. For caregivers of individuals with schizophrenia, this symptomatology occurs at a rate of 72–83% for stress [77].

Consistent with the recent findings of researchers [78], the coefficients for the variables stress, anxiety, and depression were found to have a negative sign, indicating that their eventual increase causes a decrease in the quality of life of caregivers of people with mental disorders. (RQ3) Several researchers have extensively examined the burden of informal caregivers due to the manifestation of anxiety symptoms, confirming the present study’s findings. Caregivers must provide care for an adult relative with a mental disorder, but they need assistance to do so adequately. Providing informal caregivers of individuals with mental disorders with support services can improve the caregivers’ physical and mental health and the patients’ through better management.

The impact of caregivers on the quality of life of individuals with mental disorders is substantial. Numerous studies have examined the correlation between caregiving and the prevalence of clinical manifestations and symptoms among individuals providing care for individuals with mental disorders. Researchers [79] conducted a comprehensive review and meta-analysis to investigate the extent of care responsibilities experienced by caregivers of Iranian individuals with chronic illnesses. The research revealed that individuals caring for patients with mental disorders, such as Alzheimer’s, encountered a more significant caregiving burden. The responsibility of providing care was correlated with a heightened susceptibility to mental disorders among individuals fulfilling the role of caregivers [79]. In England, a study was conducted by researchers [80] to examine the prevalence of mental and physical illness among caregivers. According to the same researchers [80], the research revealed that individuals in the role of caregivers exhibited elevated levels of psychiatric symptomatology compared to those not in a caregiving role. According to the same research [80], there is a correlation between the extent of caregiving and negative impacts on mental health, as well as an increase in psychiatric symptoms. The research conducted in [81] investigated the firsthand accounts of caregivers regarding individuals with severe mental disorders residing in rural areas of Ghana. According to this research [81], caregivers encountered a range of difficulties, such as managing the indications and manifestations of mental disorders, shouldering emotional burdens, facing instances of violence, adapting to changing roles, confronting societal stigma, and navigating disrupted family dynamics [82]. The researchers [83] conducted a study to examine the caregiving responsibilities faced by families of individuals diagnosed with schizophrenia. According to previous research [83], caregivers encountered a noteworthy burden, resulting in detrimental effects on their mental well-being. Also, the same researchers [84] proposed the implementation of family interventions and psychosocial support as potential strategies to tackle the challenges mentioned above effectively. The burden experienced by caregivers of individuals diagnosed with bipolar disorder was examined in a study conducted by researchers [85]. According to this research [85], a high caregiver burden was revealed, significantly impacting the well-being of patients and caregivers. The researchers [85] underscored the significance of acknowledging and addressing the burden experienced by caregivers within clinical and psychosocial interventions for individuals with bipolar disorder [86].

In their study [52], researchers investigated the extent of familial burden experienced by individuals providing care for patients diagnosed with schizophrenia. The research conducted by the same researchers [52] revealed a substantial occurrence of caregiver burden, which correlated with various socio-demographic factors. Researchers in this study [52] emphasized the necessity of implementing comprehensive interventions to alleviate the burden experienced by caregivers. In a qualitative investigation [87] in Saudi Arabia, caregivers expressed encountering a range of difficulties, encompassing the management of indicators and manifestations of mental disorders, emotional strain, instances of violence, alterations in familial roles, societal stigma, and disruptions in family dynamics [87]. The challenges mentioned above have played a role in exacerbating the difficulties individuals face in the role of caregivers [87]. In general, these studies underscore caregiving’s substantial influence on the overall well-being of individuals who care for individuals with mental disorders. Caregivers frequently encounter heightened psychiatric symptomatology, burden of care, and a range of challenges that have the potential to impact their mental well-being. The importance of addressing the caregiver burden and implementing suitable support and interventions cannot be overstated in terms of enhancing the welfare of caregivers and the standard of care delivered to individuals with mental disorders.

Numerous research studies have been conducted to explore this association, yielding consistently congruent findings. Other researchers [88] conducted a meta-analysis wherein they examined the relationship between levels of depressive and anxiety symptoms in caregivers and their quality of life. The findings of the study indicated that caregivers with elevated levels of depressive and anxiety symptoms experienced a diminished quality of life. In another study [89], it was discovered that caregivers who experienced elevated levels of anxiety and depression exhibited a diminished health-related quality of life. The study [27] employed a cross-sectional design to examine the relationship between anxiety and depression levels in patient–caregiver dyads. The study’s findings revealed a noteworthy correlation between these psychological symptoms, further linked to a diminished quality of life. The association between symptoms of depression and anxiety and the overall quality of life has been documented in specific populations of caregivers as well. An investigation [90] revealed that caregivers of individuals diagnosed with schizophrenia who exhibited elevated levels of depressive symptoms also reported diminished family functioning and a reduced quality of life. The examination of the relationship between symptoms of depression and anxiety and the overall quality of life has also been explored within the framework of particular conditions. Another study [91] revealed that caregivers of individuals with mental illness who exhibited elevated levels of depressive symptoms experienced diminished vitality and poorer overall health, both of which are integral aspects of quality of life. In a separate investigation [92], it was discovered that the association between neuropsychiatric symptoms in individuals with dementia and caregiver mental health was mediated by caregiver burden and affiliate stigma. Furthermore, the impact on quality of life was subsequently influenced by the caregiver’s mental health. Caregivers who experience higher levels of these symptoms tend to have a diminished quality of life [93,94,95].

The present Investigation endeavored to communicate with other researchers on a related topic. In this context, the possible absence of similar research on caregivers of individuals with a mental disorder prevented the comparison of the results with other findings of a similar nature in order to identify any similarities or differences in the sample’s attitudes towards the subject of the study. It has also been a deficiency that comparable surveys of caregivers for chronically ill patients have utilized various research methods. In addition, the sample size is limited to a single institution in the country (General Institution), preventing the generalization of the results. Indirectly indicating the quality of life of caregivers of individuals with mental disorders, the results of this study play a crucial role in supporting development policies. All these findings highlight the importance of addressing the mental health needs of caregivers and providing appropriate support to improve their overall well-being and quality of life.

Notwithstanding its limitations, such as restricted sample size and confinement to a single institutional setting, which may impede its generalizability, this study provides valuable perspectives on comprehending the burden experienced by caregivers. The statement effectively communicates the importance of prioritizing the well-being of caregivers to enhance patient management. Moreover, this study promotes the need for ongoing research in this crucial domain, intending to implement focused approaches that directly influence the well-being of caregivers.

An additional limitation of the current research project was the absence of a control group integration. By conducting separate analyses of subgroups, one can gain significant insights into discrete demographic cohorts. Transparency is essential for the precise interpretation of the findings.

Caregivers of individuals with mental disorders are significantly burdened, and problems arise in all aspects of their lives. Multiple studies have documented the high burden and its association with low quality of life, and the present study confirms this association. This burden is anticipated to increase if we consider the factors that have considerably impacted families in recent years, such as the COVID-19 pandemic and the ongoing economic crisis in our country. It is regarded as necessary to provide social and financial support and develop strategic treatment programs to manage patients better and improve their overall health [96,97,98].

Future work stemming from this research should aim to address the underlying causes of the stress, anxiety, and depression identified among caregivers, potentially by implementing and evaluating intervention programs focused on support and stress reduction. This could include more in-depth investigations examining how different types of support (psychological, financial, community based, etc.) affect caregiver well-being [99,100,101,102,103]. Moreover, a longitudinal perspective could provide insight into how caregiving affects mental health over time and if and how this impact changes as the patient’s condition evolves [104,105,106,107,108,109]. It is important to emphasize the need for a specific national strategic plan to be established for providing assistance to caregivers in Greece. In recent years, there has been a strong emphasis on the use of psychoeducation by primary healthcare facilities, both public and private mental health centers, for the most part.

Additionally, the research could be broadened to look at populations in various geographic and cultural contexts to account for differing healthcare systems, social support structures, and caregiving norms, which could influence caregivers’ quality of life and mental health [110,111,112,113,114]. There is also the opportunity to use mixed methods approaches by incorporating qualitative research, which can provide a richer, more nuanced understanding of the caregivers’ experiences [115].

## 6. Conclusions

In conclusion, this research paper underscores the profound and complex burden that caregiving for individuals with mental disorders imposes on caregivers. This study’s analysis, drawn from 157 caregivers, reveals a discernible deterioration in the quality of life for these caregivers, punctuated by prominent symptoms of stress, anxiety, and depression. It highlights the intrinsic link between the mental health of the caregiver and demographic factors such as gender, age, educational level, and employment. Notably, the study showcases that female caregivers are particularly vulnerable to the psychological distress associated with caregiving.

The findings of this research articulate the urgent need for targeted support interventions. The correlation between poorer quality of life and increased levels of stress, anxiety, and depression suggests that strategic treatment programs, along with comprehensive social and financial support systems, are essential. These interventions are vital not only for ameliorating the mental health of caregivers but also for ensuring the sustained, effective care of individuals with mental disorders.

Additionally, the exacerbation of challenges posed by the COVID-19 pandemic has only intensified the necessity for such support. Future research is imperative to optimize and tailor these support systems to the unique needs of mental disorder caregivers. Through continued investigation and the implementation of robust support measures, we can hope to improve the lives of both caregivers and their loved ones.

Therefore, while caregivers undertake their crucial roles with resilience and determination, the responsibility falls to healthcare systems, policymakers, and the broader community to recognize and act upon the silent trials they endure. We should ensure that these unsung heroes receive the comprehensive assistance they need to maintain the well-being of those they care for and their own health and quality of life.

## Figures and Tables

**Table 1 healthcare-12-00269-t001:** Demographics.

Demographics	Categories	Ν	f%
Sex	Male	48	30.6
	Female	108	68.8
	Other	1	0.6
Age	Below 25 years old	5	3.2
	26–35 years old	22	14.0
	36–45 years old	40	25.5
	46–55 years old	53	33.8
	Above 55 years old	37	23.6
Nationality	Greek	155	98.7
	European Union	2	1.3
Family Status	Single	40	25.5
	Married—Cohabitation Agreement	94	59.9
	Divorced—Termination of Cohabitation Agreement	15	9.6
	Widowed	8	5.1
Educational Level	Illiterate	1	0.6
	Primary Graduate	28	17.8
	Secondary Graduate	40	25.5
	Bachelor	59	37.6
	Master	27	17.2
	Ph.D.	2	1.3
Job Occupation	Unemployed	7	4.5
	Household	20	12.7
	Self-employed	10	6.5
	Private Employee	63	40.1
	State Employee	39	24.8
	Retired	18	11.5
Income	EUR 0–11,000	41	26.1
	EUR 11,001–13,000	36	22.9
	EUR 13,001–23,000	51	32.5
	EUR > 23,001	15	9.6
	I do not wish to answer	14	8.9
Years of Caregiving	0–10 years	116	73.9
	11–20 years	19	12.1
	21–30 years	17	10.8
	30 years and above	5	3.2

**Table 2 healthcare-12-00269-t002:** Descriptive statistics of SF-12 questionnaire questions.

Factors	Mean	SD	Median	Frequency				
				1	2	3	4	5
Moderate intensity activities (PF)	2.57	0.653	3.00	8.9	25.5	65.6	-	-
Climbing a few flights of stairs (PF)	2.54	0.615	3.00	6.4	33.8	59.9	-	-
Complete fewer tasks (RP)	3.76	1.123	4.00	3.8	10.8	22.3	31.8	31.2
Restricted work (RP)	3.78	1.184	4.00	5.1	12.7	14.0	35.0	33.1
Pain effect (BP)	4.16	0.930	4.00	0.6	8.3	7.6	41.4	42.0
General health status (GH)	3.64	0.735	4.00	-	8.3	26.8	58.0	7.0
Activity (VT)	3.22	1.04	3.00	2.5	26.8	26.8	33.8	10.2
Social activities (SF)	3.64	1.138	4.00	3.8	14.0	23.6	31.2	27.4
Complete fewer tasks (RE)	3.50	1.269	4.00	7.00	19.7	16.6	29.9	26.8
Careless operations (RE)	3.61	1.279	4.00	7.6	16.6	12.7	33.1	29.9
Tranquility (MH)	3.16	1.04	3.00	6.4	21.0	29.9	35.7	7.0
Bad mood/melancholy (MH)	3.01	1.174	3.00	11.5	24.2	26.1	28.7	9.6

**Table 3 healthcare-12-00269-t003:** Descriptive statistics of PCS and MCS subscales.

Descriptive Statistics	PCS	MCS
Mean	48.68	35.37
Standard Deviation	8.41	10.73
Median	46.99	35.97
Min.	27.59	13.01
Max.	65.94	56.10

**Table 4 healthcare-12-00269-t004:** Independent sample *t*-test controls for Gender.

	T	Sig.
PCS	−1.154	0.251
MCS	−3.740	0.000

**Table 5 healthcare-12-00269-t005:** Kruskal–Wallis test values.

Demographic Factors	Scales	*t*	Sig.
Age	PCS	21.77	0.000
	MCS	8.17	0.086
Marital Status	PCS	9.55	0.023
	MCS	7.86	0.049
Educational Level	PCS	14.70	0.012
	MCS	15.89	0.007
Employment	PCS	32.66	0.000
	MCS	37.17	0.000
Income	PCS	14.17	0.007
	MCS	1.69	0.792

**Table 6 healthcare-12-00269-t006:** Descriptive statistics of DASS-21 questionnaire questions.

Items	Mean	Standard Deviation	Median	Frequency Response			
				0	1	2	3
I couldn’t calm myself down.	0.78	1.00	0.762	40.8	41.4	16.6	1.3
My mouth felt dry.	0.34	0.00	0.607	72.0	22.3	5.1	0.6
I could not experience a positive feeling.	0.65	1.00	0.741	49.7	36.9	12.1	1.3
I was having trouble breathing.	0.45	0.00	0.692	66.2	23.6	9.6	0.6
I found it difficult to take the initiative to do some things.	0.81	1.00	0.863	43.9	35.7	15.9	4.5
I had a tendency to overreact to the situations I was faced with.	1.23	1.00	0.854	21.0	41.4	31.2	6.4
I felt shaky.	0.39	0.00	0.695	71.3	20.4	6.4	1.9
I often felt nervous.	1.16	1.00	0.836	19.1	54.8	17.2	8.9
I worried about situations where I might panic and look foolish to others.	0.72	1.00	0.838	49.7	31.8	15.3	3.2
I felt like I had nothing to look forward to.	0.75	1.00	0.831	47.1	34.4	15.3	3.2
I found myself feeling annoyed.	1.22	1.00	0.821	17.2	51.0	24.2	7.6
It was hard for me to relax.	1.20	1.00	0.851	23.6	37.6	34.4	4.5
I felt depressed and disappointed.	0.99	1.00	0.820	29.3	46.5	19.7	4.5
I couldn’t stand anything that kept me from continuing what I was doing.	0.67	1.00	0.683	43.3	48.4	6.4	1.9
I felt very close to panic.	0.46	0.00	0.772	67.5	21.7	7.6	3.2
Nothing could make me feel excited.	0.65	0.00	0.808	51.6	36.3	7.6	4.5
I felt like I wasn’t worth much as a person.	0.58	0.00	0.878	63.1	21.0	10.8	5.1
I felt that I was quite irritable.	1.17	1.00	0.90	23.6	45.2	21.7	9.6
I could feel my heart beating without any previous physical exercise.	0.76	1.00	0.804	41.4	47.1	5.7	5.7
I felt scared for no reason.	0.57	0.00	0.745	54.8	37.6	3.8	3.8
I felt that life had no meaning.	0.25	0.00	0.598	80.9	14.6	2.5	1.9

**Table 7 healthcare-12-00269-t007:** Descriptive statistics of stress, anxiety and depression subscales.

	Stress	Anxiety	Depression
Mean	14.87	7.38	9.36
Standard deviation	9.08	7.36	8.46
Median	14.00	6.00	6.00
Min.	0.00	0.00	0.00
Max.	40.00	38.00	42.00

**Table 8 healthcare-12-00269-t008:** Independent sample *t*-test controls for Gender (MCS).

MCS	*t*	Sig.
Stress	0.66	0.512
Anxiety	5.51	0.000
Depression	2.80	0.006

**Table 9 healthcare-12-00269-t009:** Kruskal–Wallis test values demographics statistical analysis.

Demographic Factors	MCS	*t*	Sig.
Age	Stress	7.25	0.123
	Anxiety	13.44	0.009
	Depression	13.95	0.007
Marital Status	Stress	15.37	0.002
	Anxiety	26.54	0.000
	Depression	21.82	0.000
Educational Level	Stress	19.57	0.002
	Anxiety	26.57	0.000
	Depression	21.71	0.001
Job Occupation	Stress	23.48	0.000
	Anxiety	10.92	0.053
	Depression	6.89	0.229
Income	Stress	5.30	0.258
	Anxiety	7.54	0.110
	Depression	2.66	0.617

**Table 10 healthcare-12-00269-t010:** Pearson’s correlation coefficient.

	PCS	MCS	Stress	Anxiety
MCS	0.056			
Stress	−0.212 **	−0.596 **		
Anxiety	−0.200 *	−0.457 **	0.697 **	
Depression	−0.224 **	−0.580 **	0.759 **	0.758 **

* Statistically significant association at the 5% level; ** statistically significant association at the 1% level.

## Data Availability

Data available on request due to privacy/ethical restrictions.

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
