# Peer review of "Quality of Life and Incidence of Clinical Signs and Symptoms among Caregivers of Persons with Mental Disorders: A Cross-Sectional Study"

_healthcare, 2024, doi:10.3390/healthcare12020269_

Round 1

Reviewer 1 Report

Comments and Suggestions for Authors

It is an interesting research among caregivers to individuals with mental disorders. It's a very meaningful topic because mental disorders are unique conditions that can put heavy tolls on their caregivers' well-being.

Research questions and study endpoints are clearly defined, and a thorough literature review was given.   However, during review of method and results sessions, I have identified several potential issues below: 

1. how to quantify exposure? The exposure is "giving care to indiviudals with mental issues". However, those patients visited a psychiatric departiemtn have different levels and types of disorder, which requires various levels of care. So the exposure can be measured in duration of care and intensity of disorder, which adds lots of variability to the study sample.

2. the research question #2 states which demographic factors significantly influence the level of QoL of caregivers. The term "influence" indicates causal direction. It is more likely an association, unless you can use a randomized trial or manipulate that factor to observe before/after condition. However, it is a cross-sectional one time point analysis. It addresses associations, but "influsing factor" is a bit difficult to explore. Also, those findings might be found in general population, such as income vs PCS, employment vs PCS/MCS. We'll need to compare general population to findings in this study population to draw a population-specific conclusion.

3. correlation between depression/anxiety and QoL. I understand there must be a correlation found. But the study focuses on caregivers for individuals with mental disorders. Do we have data in Greece's general population on the same topic to see if that correlation is different in this study's target population?

4. do we know if the caregiver is a close relative, or a paid worker? I believe that would be a major influencing factor as well.

5. I am not sure what Figure 1 is trying to prove in the manuscript. It looks a little out of place.

6. Please also note the alpha level of 0.05 is not under multiplicity control. Therefore they're only nominal, and descriptive in nature.

7. How is extremely imbalanced demographics handled? for instance, only 3.2% < 25 years old, only 2 EU participants, only 1 illiterate, 2 PhD, etc.

8. How are missing responses handled?

9. Analyses in Table 3 is confusing: correlation between individual factors and their sums? Since we know the questionnaire has good internal consistency via Cronbach's index, this analysis does not really make much sense to me personally. It would make sense to conduct a factor analysis or SEM to compare factor loadings, but that's only suggestion.

10. watch out for statements such as "the mean for samples greater than 30 follows the normal distribution". You may claim it is a practically close approximation, but asymptotic properties shall be stated with caution.

11. it is alarming that the rationale authors picked Kruskal-Wallis test over ANOVA is "the analysis would not produce statistically significant results". It is a statement of p hacking. You can always combine categories in a way that makes clinically sense, or pick the non-parametric version due to concerns of residual distribution. However, "Kruskal-Wallis test can extract statistically significant results" is never an acceptable rationale. I think the author is trying to say "Kruskal-Wallis test is more powerful to detect differences in this data pattern."

12. A subgroup analysis such as forest plot would help with presentation. We see the Kruskal-Wallis test results, but it does not give a direction.

13. Regression in Table 9 is not a well designed model. We know the 3 mental factors (stress, anxiety, depression) are correlated themselves (and also according to Table 8). Putting them into one model will suffer from model multi-collinearity.    I would like to see those points addressed to make it a more solid statistical report.

Author Response

Dear Reviewer,

Wishing you a Prosperous New Year 2024!

Thank you for your valuable comments. Addressing your questions, we would like to inform you about the following:

1 & 2 have been corrected

3. There are no data on Greece's general population to correlate them with our study's target group. It is a nice idea to for further research after the present study.

4. All the participants in the study were close relatives from local regions nearby Peloponnisos in Greece.

5. Thank you for the comment regarding Figure 1, we checked and eliminated it.

6. It is corrected.

7. Concerning imbalanced demographics, we analyzed subgroups separately. Also we clearly state this as a limitation in our reporting in discussion.

8. The missing values are omitted, and the analysis is conducted with the remaining available data.

9. Table 3 is deleted. Thank you for the comment.

10-13. All the questions addressed are corrected and some allow us to consider better the usefulness of these themes (for instance Regression in Table 9) which is deleted.

In the manuscript, the changes are highlighted in green color.

Thank you again for the constructive collaboration

On behalf of the authorship

Evgenia Gkintoni

Reviewer 2 Report

Comments and Suggestions for Authors The aim of the reviewed article was to assess the quality of life and selected mental health problems (stress, depression, anxiety) of people who are caregivers of people suffering from mental disorders. The authors comprehensively justified the selected topic and described the research methodology. The results were presented in a comprehensive manner, and the selection of literature was substantively justified.   It should be noted that the issue of quality of life and health problems of caregivers of mentally ill (chronically ill) people is not a topic often discussed in scientific works. The vast majority of research concerns disorders occurring in caregivers of chronically ill people. Due to the observed increase in the number of mental disorders and their impact on family functioning, the topics discussed by the authors are particularly important.  

I propose to briefly present whether the medical care system in Greece supports caregivers of mentally ill people in any way. Taking into account the importance of the topic discussed by the authors, it seems reasonable to establish international cooperation and continue research among people living in other areas of Europe.

Author Response

Dear Reviewer,

Wishing you a Prosperous New Year 2024!

We appreciate the insightful feedback you have provided. A distinct national strategic framework needs to be present regarding the provision of support for carers in Greece. The endorsed policy emphasizes the utilization of psychoeducation by primary healthcare facilities, including public and private mental health centers, for the most part.

Promoting international collaboration and ongoing research among individuals residing in different European countries is essential for effectively managing the issue under discussion. Our research team is inclined to engage in academic collaboration with other scholars, where we can exchange perspectives and scientific expertise and implement research protocols that yield tangible advantages for both local and global communities. Also, the suggestion about the presentation of the Greek medical care system in caregivers’ support is added in the discussion as indicated (lines 680-683), and the limitation about the control group (lines 661-664).

On behalf of the authorship

Evgenia Gkintoni

Reviewer 3 Report

Comments and Suggestions for Authors

1. Introduction

This is a genuine piece of empirical research that attempts to examine a topic which has been investigated extensively in the international literature facing thus, from the outset, a challenge to persuade the audience what novel contribution brings to the corpus of knowledge. 

In this perspective, the authors acknowledge the extensive body of evidence in caregivers' quality of life (Line 45) and they declare they would focus on mental health caregivers.  However the authors fail to acknowledge the body of evidence on the particular aspect, which is also extensive.

In relation to the above observation, the subsequent paragraph (from L52) confirms the narrow focus of this work, which disregarding the international literature, rushes to introduce the reader to the country where the data were collected (i.e. Greece). Although it is important to describe explicitly the context, this cannot be accomplished at the expense of the broader international context and the literature. In this respect I would suggest to revise according the first 3 paragraphs.

Up to the end of this introductory section, the authors have not made clear to the audience whether they will focus on informal caregivers (such as  family and the community) or professional caregivers, causing thus confusing. This has to be considered and provide the necessary clarification.

2. Literature review 

This section is well-structured, although a number of language errors may jeopardise the good work. An indication is cited below:

Everyone's interpretation of the term quality of life is highly subjective. (L74): This a big statement unsupported by evidence. 

In the social sector (L79): The authors most likely do not actually refer to "sector" as social sector refers to social businesses, political and religious organisations, community groups and other institutions. They may refer to social dimension or aspect of the quality of life.

it has been identified as positive predictors of the quality of life (L99): syntax error

"Unfortunately (L119)": I suggest you avoid references to luck in scientific texts.

Leaving aside the language problems, I wish to emphasise my concern how is the literature been used in this manuscript. For instance, in Line 105 the authors refer to "melancholy". I have checked the reference (i.e. 13) in BMC Psychiatry and I have not traced it in the text. I am ambivalent whether this term is in use anyway.  

"Caregivers of people with mental disorders must be supported formally and informally by family members, relatives, friends, and the nation's health policy to maintain their health and well-being." (L117-118) This is a highly problematic statement in that it shoves the responsibility to family and the community to train caregivers!

It is also highly problematic the fact that until Line 118 the authors have not made clear to the kind of caregiving they investigate.

In line 119 the authors finally refer to informal caregiving without making clear if this constitutes the focus of their work.

From line 140 to 149 a paragraph referring arbitrarily to patients makes no sense. How does this paragraph relate to the rest of the text and most essentially with this research?

3. Materials and methods

Symptomatology (also called semiology) is a branch of medicine dealing with the signs and symptoms of a disease. Are the authors sure that this what they mean by this term in line 248?

Has DASS-21 been standardised before this research?

4. Discussion

Usually the authors attempt to synthesise and discuss the produced evidence against the literature building their own argumentation, often leading to a thesis. This was hardly the case in terms of this discussion.

For instance, from L449-457 the authors reiterate percentages from a survey conducted by EUROFAM CARE (?) without linking it to their work, without even citing a reference for this.

In L 455  the authors mention "Some studies attribute a more significant burden to elderly caregivers". First it is not proper academic language to use noun that refer to indefinite quantities, since science has to be accurate and precise, and second there is not even a reference to support the claim.

 What does this [RQ1] stand for in this section?

"the findings of Dimitropoulos (2022) L 489 ": this is not acceptable within this reference style. Also, and this is highly problematic, the authors refer to the findings of Dimitropoulos (2022) and referring to [49] and [51] citations in the end, which do not correspond to the reference. The surname is not there!

The whole of this section needs to be more robust, with less wording but mor concrete argumentation consistent with the findings.

Comments on the Quality of English Language

Please see above for details.

Author Response

Dear Reviewer,

Wishing you a Prosperous New Year 2024!

Thank you for your valuable comments. We took into account all your remarks in the sections of the manuscript and corrected them in the lines as indicated. Some phrases that were considered as non-appropriate are deleted and replaced according to your indications. The revisions are highlighted in green color.

Here are presented point by point your indications for revision:

Introduction:

In lines 52-90 addressing your comments, we added further information and data not only for Greece but also for a broader international context. All changes are highlighted in green color.

Literature review:

Lines 107-114, 131-138, 149-156, and 170-179 are revised and have been considered all the changes as indicated. Some phrases that were considered non-appropriate are deleted and replaced according to your indications. For instance, phrases like unfortunately, (L119) have been deleted, also problematic statements such as melancholy replaced with depression or other statements "Caregivers of people with mental disorders must be supported formally and informally by family members, relatives, friends, and the nation's health policy to maintain their health and well-being” or statements in lines 119 and 140-149 have been also deleted. The revisions are highlighted in green color.

Materials and Methods:

Symptomatology in line 248 replaced to symptoms with a more general and non-clinical term

Lines: 261-268 There is an addition for the standardization of the scale DASS

Discussion:

References non appropriate like in L449-457 or L455 are deleted as indicated. Additionally, reference to RQ1 is deleted and reference in L489 is also omitted as indicated.

Thank you again for our constructive collaboration and for the time you devoted to the amelioration of our manuscript.

On behalf of the authorship

Evgenia Gkintoni

Reviewer 4 Report

Comments and Suggestions for Authors

This article provides valuable insights into the psychological health challenges faced by caregivers. The structure of the paper is logical, and the research design and methodology are clearly articulated. However, the paper could be further strengthened by adding a more detailed discussion of existing studies, more specifically justifying the choice of methods, and making a deeper comparison of the research findings with existing literature. Finally, the discussion section could be expanded to include more specific policy recommendations and practical guidance, enhancing the research's applicability and impact.

- In the literature review, it is suggested to highlight the gaps in existing literature regarding the quantification of caregiver burden and its impact on quality of life, and how this study addresses these gaps. Emphasize the significance and innovation of this paper.

- In the methods section, it is recommended to explain why SF-12 and DASS-21 were chosen as assessment tools. What advantages do they offer for your research compared to other assessment tools?

- In the results section, it is advised to specifically compare and discuss how the study's findings differ from previous research, such as "Compared to the findings of study [8], we discovered that the quality of life of caregivers is closely related to their marital status and educational level."

Overall, this is a high-quality study that, with some minor adjustments, could contribute significantly to academic research and have substantial practical application potential.

Author Response

Dear Reviewer,

Wishing you a Prosperous New Year 2024!

Thank you for your valuable comments. The revisions are highlighted in green color.

Thank you again for our constructive collaboration and for the time you devoted to the amelioration of our manuscript.

On behalf of the authorship

Evgenia Gkintoni

Round 2

Reviewer 1 Report

Comments and Suggestions for Authors

Comments have been addressed. There's room to improve from a statistics and study design perspective, especially to compare to the general population. It may not be a solid conclusion if similar conclusions can be found in the general population. Also, we do need to quantify the level of mental disorder of the patients, because that for sure will pose different levels of stress on their care-givers. But given it's not a statistics-centered paper, I would be okay since most stat-related comments are addressed.

Author Response

Thank you for your email and your valuable comments for the amelioration of our manuscript. Addressing the themes as indicated, please find attached our manuscript revised. All the changes are highlighted in green color. Some extra clarifications regarding methods, informed consent/ ethical approval are added.
On behalf of the authorship

Evgenia Gkintoni

Reviewer 3 Report

Comments and Suggestions for Authors

The review has been considered careful and

an amended  version is now submitted. 

Author Response

Thank you for your email and your time devoted to the amelioration of our manuscript. Some extra clarifications regarding methods, informed consent/ ethical approval are added. All the changes are highlighted in green color.

On behalf of the authorship

Evgenia Gkintoni